# The Impact of Neurological Complications in Endocarditis: A Systematic Review and Meta-Analysis

**DOI:** 10.3390/jcm13237053

**Published:** 2024-11-22

**Authors:** Federico Sanguettoli, Federico Marchini, Federica Frascaro, Luca Zanarelli, Gianluca Campo, Christoph Sinning, Timothy C. Tan, Rita Pavasini

**Affiliations:** 1UO Cardiologia, Azienda Ospedaliero Universitaria di Ferrara, 44124 Ferrara, Italy; f.sanguettoli@gmail.com (F.S.); mrcfrc2@unife.it (F.M.); federica.frascaro92@gmail.com (F.F.); luca.zanarelli@edu.unife.it (L.Z.); cmpglc@unife.it (G.C.); 2Department of Cardiology, University Heart & Vascular Center Hamburg, 20246 Hamburg, Germany; c.sinning@uke.de; 3Department of Cardiology, Blacktown Hospital, Western Sydney University, Sydney 2148, Australia; timothy.tan@health.nsw.gov.au; 4Department of Cardiology, Westmead Hospital, Sydney University, Sydney 2148, Australia

**Keywords:** endocarditis, stroke, mortality, cardiac surgery

## Abstract

**Background:** Infective endocarditis (IE) is associated with significant neurological complications (NCs). The impact of neurological sequelae due to IE, however, is not well characterized. Thus, the aim of this systematic review and meta-analysis is to determine whether patients who experienced NCs from IE had worse outcomes compared to those without neurological complications. **Methods:** We conducted a systematic and comprehensive literature search of MEDLINE, Cochrane Library, Google Scholar, and BioMed Central (PROSPERO registration ID: CRD42024518651). Data on the primary outcome of all-cause mortality and the secondary outcome of surgical timing were extracted from 25 observational studies on patients with confirmed IE, both with and without NC. **Results:** In the pooled total of patients with IE, NCs were present in 23.7% (60.8% ischaemic stroke and 16.4% haemorrhagic stroke). All-cause mortality was significantly higher in patients with IE and NCs (OR 1.78, CI 1.47–2.17, *p* < 0.0001) compared to those without, particularly in those with major neurological events (OR 2.18, CI 1.53–3.10, *p* < 0.0001). Conversely, minor or asymptomatic strokes showed no significant correlation with mortality (OR 1.10, CI 0.82–1.47, *p* = 0.543). There was no significant difference in the timing of surgical intervention (standardized mean difference −0.53, CI −1.67 to 0.61, *p* = 0.359) between the two patient groups. **Conclusions:** Major NCs due to infective endocarditis were associated with a significantly increased all-cause mortality. This underscores the critical importance of early recognition and management strategies tailored to the severity of neurological events.

## 1. Introduction

Infective endocarditis (IE) is a serious condition which can carry high morbidity and mortality rates. There has been a steady increase in the prevalence of IE in recent years, due in part to the aging population, an increase in the rates of the implantation of devices, such as pacemakers and defibrillators, and prosthetic valves, and the increased use of indwelling intravenous catheters [1,2,3].

Neurological complications (NCs) can often complicate the course of IE and contribute significantly to the mortality and morbidity associated with IE [4]. These include embolic cerebrovascular complications, intracranial haemorrhage, ruptured mycotic aneurysms, transient ischaemic attacks, meningitis, encephalopathy, and brain abscess, with up to 40% reported to be symptomatic and up to 80% asymptomatic [5,6]. NCs are also one of the reasons for delayed cardiac surgical interventions and can represent a significant management challenge in a patient with IE [4]. Delaying surgical intervention in patients with IE can lead to clinical deterioration and potentially death. There is a paucity of high level evidence for surgery in the treatment of IE in the current guideline recommendations by the American College of Cardiology (ACC), American Heart Association (AHA), and European Society of Cardiology (ESC). However, current guidelines do support urgent surgical intervention in the case of uncontrolled infection, persistent sepsis, acute heart failure, or a high risk of embolic recurrence [7], especially in the case of ischaemic or haemorrhagic stroke with favourable characteristics. Otherwise, the recommendation is that surgery should be deferred for up to 4 weeks where possible [7].

The decision to proceed with surgical intervention can be complicated as the need for surgery has to frequently be balanced with the increased risk of a poorer outcome associated with the need for required anticoagulation to prevent thrombosis within the extracorporeal circuit used for cardiopulmonary bypass. In patients who have recently experienced a stroke, anticoagulation also poses a significantly increased risk of a haemorrhagic transformation of the ischaemic brain tissue, particularly since the inflammatory and septic milieu of IE also carries an added risk of potential lethal intracranial bleeding [8].

Finally, the long-term prognosis of the different neurological complications (NCs) in the context of infective endocarditis is not well characterised. While there is agreement among studies that major NCs carry a significant risk of mortality, there is a paucity of evidence with regards to minor or asymptomatic strokes due to the heterogeneity in reported patient outcomes and variability in the definitions and reporting of neurological events. Hence the aims of this review are to (i) evaluate the association between NCs in patients with IE and mortality, focusing particularly on those with minor or asymptomatic strokes, and (ii) to determine if the occurrence of NCs in patients with IE is associated with delay in the time to surgical intervention.

## 2. Materials and Methods

We performed a systematic review and meta-analysis according to the principles outlined in the Preferred Reporting Items for Systematic reviews and Meta-Analyses (PRISMA), Quality of Reporting of Meta-analyses (QUOROM) statement, and recommendations from The Cochrane Collaboration and Meta-analysis of Observational Studies in Epidemiology (MOOSE) [9]. The study protocol of the present meta-analysis has been registered with Prospero (ID: CRD42024518651).

### 2.1. Search Strategy

Appropriate articles were identified following MESH strategy in MEDLINE, Cochrane Library, Google Scholar, and BioMed Central databases. The following terms were searched: “endocarditis” AND “neurological complication” AND “mortality”. The search strategy was carried out in February 2023 and updated in June 2024. Only full-text articles, published in English and in peer review journals were selected. Two expert cardiologists (RP and FM) independently reviewed databases looking for studies regarding outcomes of patients with IE and NCs. Disagreements were resolved through discussion and consensus. In the case of unresolved disagreement, a third reviewer (GC) intervened to reach consensus. The reference list of included studies and the reference lists of relevant reviews were also reviewed for additional studies which were missed in the initial search.

#### 2.1.1. Inclusion Criteria

Studies with the following characteristics were included in the meta-analysis: (a) studies that compared adult patients (age of ≥18 years) with a confirmed diagnosis of IE with and without an NC; (b) with data on mortality in the case of NCs; (c) reporting data on the timing regarding the delay in cardiac surgery in the case of NCs; (d) reporting on the effect size of data regarding mortality as a hazard ratio (HR) or an odds ratio (OR) with 95% confidence interval (CI); or (e) the absolute number of events in each study group (patients with versus without NC).

#### 2.1.2. Exclusion Criteria

Studies were excluded based on the following criteria: (a) they were only in abstract or poster form; (b) studies on children (aged < 18 years old); (c) reviews, meta-analyses, or editorials; (d) rationale and study protocols; (e) the full-text article was not available; (f) they did not compare patients with and without NCs; or (g) they had missing data.

### 2.2. Data Extraction

Three independent reviewers (FS, FF, and LZ) extracted data from the full texts and published appendixes. The following information was collected for every included study: the year of publication, journal, number of patients included, time of the enrolment, follow-up length, source for follow-up, age, sex, cardiovascular risk factors, history of drug affection, site affected by the endocarditic process (native valve, prosthetic valve, intra-cardiac device, and catheters), presence of endocarditis complications (abscess, fistula, severe valve disease, or heart failure), surgery performed, timing of surgery (defined also as early or not early), and blood culture that was positive for staphylococcus aureus or fungi.

The primary outcome of interest was all-cause mortality at the longest available follow-up in patients with NCs and a diagnosis of IE. The secondary outcome was the standardized mean difference in time to cardiac surgery among patients with and without NCs. A sub-analysis which examined the impact of major and minor NCs on mortality and time to surgery was also performed, where a minor NC was defined as a minor stroke with limited residual neurological deficit, transient ischaemic accident, or asymptomatic stroke and a major NC was defined as a stroke leading to significant neurological deficit and disability, territorial stroke, or haemorrhagic stroke. Appendix A reports the definitions used by each single study to determine major or minor NCs with the corresponding odds ratio.

### 2.3. Internal Validity and Quality Appraisal

The quality of the included studies was appraised by two unblinded reviewers (FS and FM) and tested using pre-specified electronic forms of MINORS criteria [10]. Twelve methodological items were given a score between 0 and 2. “Not reported (0 points)”, “Reported but inadequate (1 point)”, or “Reported and adequate (2 points)”. The maximum score achievable was 24 points. Among studies included in the meta-analysis, the minimum score was 14 and the maximum score was 21 (Appendix A).

### 2.4. Data Analysis and Synthesis

Values for continuous variables were expressed as a mean and standard deviation (SD). For the studies which reported values as a median and interquartile range, values were converted to mean (standard deviation) values to allow for meta-analysis using a previously published formula [11]. Categorical variables were expressed as absolute numbers and a percentage of the total pooled population.

Considering the high likelihood of between-study variance, we used a random effect model. Statistical heterogeneity was assessed using the Cochran’s Q test and *I*^2^ statistic with a value of *I*^2^ of 0 to 25% considered as insignificant heterogeneity, 26 to 50% low heterogeneity, 51 to 75% moderate heterogeneity, and >75% high heterogeneity. The mean value of the time to surgery for patients with neurological complication was also expressed in terms of the standardized mean. The impact of the severity of the neurological event (minor versus major) on the primary endpoint was tested with sub-group analysis, using the ANOVA test. Finally, random effect meta-regression analysis was performed to assess the effect of some potential confounding factors including device involvement, gender (male), prosthesis endocarditis, right heart valve endocarditis, positive blood culture for *Staphylococcus aureus*, positive blood culture for fungi, the presence of endocarditis anatomical complications, to be treated with cardiac surgery, aortic valve endocarditis, mitral valve endocarditis, intravenous drug users, early surgery, heart failure, and the mean age of the population. Meta-regression was considered for the variable with data sourced from at least 10 studies. Publication bias was appraised by a graphical valuation of funnel plots and through Egger’s linear regression test, and Duval and Tweedie trim and fill. Prometa software 3 (Internovi, Cesena, Itay) and RevMan 5 (The Cochrane Collaboration, The Nordic Cochrane Centre, Copenhagen, Denmark) were the software used for statistical analyses.

## 3. Results

### 3.1. Search Results, Study Selection, and Patient Characteristics

A total of 649 studies were identified in the initial search. After the removal of duplicates and screening of the title and abstract, 27 of the 41 full-text articles were included based on the inclusion and exclusion criteria. Of the 27 studies included, two pairs of studies (by Heiro et al. [12,13] and Ruttmann et al. [14,15]) were conducted on the same cohort spanning through different periods of time; hence, the studies from each pair with lesser patients were excluded [13,15]. The final 25 studies were quantitatively analysed [5,12,14,16,17,18,19,20,21,22,23,24,25,26,27,28,29,30,31,32,33,34,35,36,37] (Figure 1). Of note, the study of Okazaki et al. [26] was included only for the analyses of the secondary outcome. There was a pooled total of 14509 patients, of which 3439 had at least one NC (23.7%). The mean age of the population was 61.9 ± 9.9 years, with a prevalence of male patients accounting for 64% of the total number of patients. Baseline data and patient characteristics are reported in Table 1.

Cardiac surgery was performed in 68.5% (2004) of patients with NCs and in 65.8% (6592) of patients without an NC (*p* < 0.0001). The heart valve most commonly involved in patients with an NC was the mitral valve (47.2%), followed by the aortic valve (36.9%) and then the right heart valves (4.3%). Prosthetic valve endocarditis was present in 23.2% of NC patients. Abscess or local valvular complications (namely abscess, fistula, or new severe valve disease) were present in 26.2% of the patients with NCs. There was no significant difference in the rates of heart failure on admission to hospital in those with an NC vs. those without (32.5% vs. 33.4%; *p* = 0.590). Staphylococcus aureus was the most frequently identified organism. Interestingly, Staphylococcus aureus was also found to be more prevalent in patients with an NC vs. those without (31.4% of patients with NC vs. 21.7% of those without; *p* < 0.0001).

Twenty-one out of twenty-seven studies reported on the different types of NCs (Appendix A). The majority of NCs were ischaemic strokes (60.8%; 2091/3439 events), followed by subclinical brain embolization or transient ischaemic attacks (4.1%; 140/3439 events), intracranial haemorrhages (16.4%; 564/3439 events), brain abscesses (1.7%; 60/3439 events), meningitis (5%; 171/3439 events), and mycotic aneurysm (0.7%; 25/3439 events). Some patients experienced two concurrent complications: 0.6% (19/3439) had an ischaemic stroke and brain abscess or meningitis, 1.4% (51/3439) had an ischaemic stroke and an intracranial haemorrhage, and 0.4% (15/3439) had a brain abscess and meningitis.

Twelve studies reported on mean surgical times defined as the time between the time of diagnosis of the onset of NCs and cardiac surgery [5,14,17,18,21,26,31,32,33,34,35,36]. In eight of these studies, the mean surgical waiting time of the control group was also reported (Table 2).

### 3.2. Primary Endpoint

All-cause mortality was significantly higher in patients with infective endocarditis with an NC (OR 1.78, 95% CI 1.47–2.17, *p* < 0.0001, *I^2^*: 75%) (Figure 2). Those who experienced a major neurological event, defined as a territorial infarct, haemorrhagic stroke, or a stroke with impaired neurological status, had an even stronger correlation with all-cause mortality (OR 2.18, 95% CI 1.53–3.10, *p* < 0.0001, *I^2^*: 83%) (Figure 3), compared to a silent or minor stroke that was not correlated with all-cause mortality (OR 1.09, 95% CI 0.81–1.47, *p* = 0.56, *I^2^*: 17%). ANOVA testing revealed a significant difference in mortality among subgroups of minor versus major NCs (*p* = 0.003) (Figure 3).

Finally, random effect meta-regression demonstrated no significant interaction between the increased risk of all-cause mortality associated with NCs and the following variables: male sex (β = −0.01, *p* = 0.450), prosthesis involvement (β = −0.00, *p* = 0.854), right heart valve involvement (β = 0.02, *p* = 0.645), S. aureus positive blood cultures (β = −0.00, *p* = 0.675), the presence of abscess (β = 0.01, *p* = 0.629), having cardiac surgery performed (β = −0.01, *p* = 0.330), aortic valve involvement (β = −0.01, *p* = 0.073), mitral valve involvement (β = 0.01, *p* = 0.124), and heart failure at admission (β = 0.00, *p* = 0.937) (Appendix A). On the contrary, a significant interaction was found with mean age (β = −0.02, *p* = 0.003). Due to the low number of studies reporting data for those with fungus infection and intravenous drug use, meta-regression analyses were not performed for these patients (Appendix A).

### 3.3. Secondary Endpoint

The meta-analysis did not demonstrate a significant standardized mean difference (SMD) in the time interval to cardiac surgery between patients with and without NCs during infective endocarditis (−0.53, 95% CI interval −1.67 to −0.61, *p* value = 0.359). The non-standardized mean difference expressed in days was −1.86 (95% CI interval −6.44–2.72, *p* = 0.426).

### 3.4. Publication Bias of the Primary Endpoint

The publication bias analysis evaluated using Egger’s linear regression test was negative (*p* = 0.101). Trim and Fill analysis indicated the potential absence of six studies; however, even with the virtual addition of these studies, the estimated effect size remained significant (OR 1.59, 95% CI 1.24–2.04, *p* < 0.0001) (Appendix A).

## 4. Discussion

NCs are the most common extracardiac complications of IE and include a heterogeneous group of conditions, the most common of which is ischaemic stroke, followed by haemorrhagic stroke, mycotic aneurysm, cerebral abscess, or encephalitis. A significant number of these NCs may occur sub-clinically or are asymptomatic [5].

The main findings of the present systematic review and meta-analysis are as follows:

(i)A rate of NCs in patients with endocarditis of 24%;(ii)All-cause mortality was significantly higher in the patients with IE experiencing any NC vs. those without an NC (OR 1.78, 95% CI 1.47–2.17, *p* < 0.0001);(iii)Major stroke or haemorrhagic stroke were related to all-cause mortality, whereas minor strokes/TIA or asymptomatic strokes were not;(iv)The presence of an NC was not significantly associated with the time to surgery among patients with and without NCs.

The rates of NCs in patients with IE in our study are consistent with the rates previously reported [8,15], although the prevalence of asymptomatic complications or transient ischaemic attacks was significantly lower in our study (4.1%). This may be an underestimation of the true prevalence of minor NCs as no distinction was made between symptomatic and asymptomatic ischaemic strokes in a significant number of studies as in many studies the diagnosis of an NC was made clinically with no brain imaging performed (Table 2).

The high prevalence of mitral valve endocarditis observed in the meta-analysis is also consistent with the reported prevalence for mitral valve endocarditis, which identifies mitral valve endocarditis as a risk factor for stroke [38,39]. Similarly, we found that *Staphylococcus aureus* was not only the most frequently isolated pathogen in subjects with infective endocarditis but also more prevalent in those with NCs. Several studies have reported a significant correlation between *Staphylococcus aureus* endocarditis and NCs [40].

The significant association between NCs and all-cause mortality aligns with multiple studies in the literature, both retrospective and prospective, where NCs have emerged as independent risk factors for short- and medium-term all-cause mortality [13,19,41,42,43]. In the European registry EURO-ENDO by Habib et al. [41], a cerebral complication was reported to have a HR of 2.21 (95% CI 1.61–3.04, *p* < 0.0001). It is important to note that the study by Habib and colleagues aimed to provide predictors of in-hospital or short-term mortality related to IE rather than focusing solely on NCs.

On the other hand, our study is unique since we have focused only on studies reporting on patients with NCs, which allowed us to identify independent risk factors associated with NCs and mortality in these patients. Our meta-regression analysis identified age as the significant primary risk factor with mortality. This inverse association between age and mortality is likely to be explained by the selection bias characteristic of this type of study, where very elderly patients with multiple comorbidities are usually excluded as inferred by the mean age of our pooled cohort that was only 61.9 ± 9.9 years. The other key question relates to the relationship between all-cause mortality and the severity of the NC. The cohort that was the focus of our study comprises a heterogeneous group of clinical conditions classified predominantly as an ischaemic stroke, which included functionally debilitating major strokes to clinically mild/minor strokes, TIAs, completely asymptomatic strokes, and other secondary complications with distinct prognostic implications such as intraparenchymal haemorrhage or cerebral abscess, which can occur independently or exacerbate the severity of an ischaemic stroke. Major or haemorrhagic strokes have been documented to confer a worse prognosis [44] and are frequently the main reason for exclusion from surgery. In the study by Garcia-Cabrera [19], patients with NCs had a mortality rate almost doubled in comparison to those without NCs (45% vs. 24%), with a cerebral haemorrhage and/or moderate–severe ischaemic event found to be independently associated with mortality. Likewise, in this study, fewer patients with NCs underwent surgery compared to patients without NCs (32% vs. 41%, *p* < 0.01) with the reasons for exclusion from surgery attributed to deteriorating neurological conditions or an unacceptably high risk of a poor outcome.

However, the impact of minor or asymptomatic strokes on the overall prognosis remains unclear. The definition of major versus minor stroke varied across studies (Table 2 and Appendix A). Our meta-analysis did not demonstrate any significant association between minor strokes/TIA or asymptomatic strokes and all-cause mortality, while major and haemorrhagic stroke were highly associated with an increased risk of all-cause mortality. This finding was consistent with the results reported in the study by Lee and colleagues [24], where major cerebrovascular events were associated with a high risk of mortality although total cerebrovascular events were not significantly related to the risk of death. In contrast, the study by Misfeld and colleagues [25], which was a surgical series of consecutive patients with endocarditis and neurological complications, found that long-term survival after surgery was not influenced by pre-operative symptomatology, but by the presence of cerebral embolism as a whole. In the study, both patients with silent or symptomatic cerebral embolism had an equally reduced long-term mortality [25].

From the published studies that we had included in our systematic review, we identified that the main surgery contraindications were haemorrhagic stroke, poor functional capacity after ischaemic stroke, or major prior comorbidities.

In these instances, excluding them from surgery effectively denies them a potentially curative therapy, resulting in the progression of infections, a worsening of local complications, and a higher risk of systemic embolisms. In the study by Arregle et al. [16], 94 out of 351 patients with IE had NCs. Even though 6-month mortality rates did not differ between the two groups, the 40 patients in which cardiac surgery was postponed due to surgical contraindication had the worst prognosis (death, new neurological event, or cardiac or septic deterioration). Among predictive factors of death in patients presenting with NCs, there was temporary surgical neurological contraindication. In the study by Thuny et al. [34], 59% of patients treated with medical therapy had a theoretical indication for surgery, whereby almost 50% died at the 1-year follow-up mark. However, when compared with patients who underwent surgery, those who had an indication for surgery but did not have surgery had more comorbidities (*p* = 0.007) and a lower Glasgow Coma Scale (14.6 ± 1 vs. 12 ± 4, *p* = 0.0001) when compared to those who underwent surgery.

Nonetheless, stroke remains a reason for delaying surgery in clinical practice, although not an absolute contraindication unless the neurological prognosis is very poor or in the case of a recent cerebral haemorrhage [6]. In our study, mean surgical waiting times varied greatly between the included studies. Most recent studies and meta-analyses in patients with IE have focused on determining whether early surgery is feasible and safe in cases of ischaemic stroke and on identifying the most appropriate waiting times for haemorrhagic stroke [40,44,45]. In our study, we showed that there was no significant difference in the mean time to surgery in patients with infective endocarditis complicated by an NC compared to those without NCs. However, several factors have to be noted: (i) Although our analysis included all NCs, a selection bias is inherent in the retrospective nature of most of the studies we considered. Patients excluded from surgery, namely the most critically ill or those with worse complications, were not likely to have been included in the analysis. (ii) Ischaemic stroke, particularly according to recent evidence, is also a factor that favours early surgery, as these patients are at an increased risk of systemic embolism and early mortality. (iii) Some of the studies considered [14,35] were specifically designed to demonstrate the safety of early surgery in these patients. Given the limited number of cases available for analysis, new studies focusing on this specific question are clearly still needed.

Finally, although cardiac surgery can be lifesaving, it also carries significant risks, which are mainly related to the need for high-dose systemic anticoagulation during cardiopulmonary bypass, which carries the risks of further bleeding, particularly in patients with intracranial haemorrhage, and potential further neurologic deterioration in those with an ischaemic stroke [44,45,46]. In our study, it is worth noting that the mean peri-operative mortality in patients with NCs undergoing cardiac surgery was 17.1% in the studies we evaluated. In one study [14], early surgery seemed safe, without a worsening of neurological symptoms in patients with ischaemic stroke. However, there was a trend towards higher perioperative mortality in patients with complicated neurological injuries (nine patients, 21.4%) compared to patients with uncomplicated ischaemic lesions (six patients, 6.5%; *p* = 0.063).

### Study Limitations

This meta-analysis has intrinsic limitations inherent to study-level meta-analyses of observational series. One major limitation is the risk of methodological heterogeneity among the included studies, which can arise from differences in study design, patient populations, diagnostic criteria, treatment protocols, and regarding the outcomes and analysis performed in particular, and as already pointed out, a great source of heterogeneity is present in the definition of major versus minor NCs as well as in mean surgical waiting times across the studies included. Additionally, treatment allocation bias or selection bias is likely present, particularly when comparing patients with different operative risks. Patients with NCs who underwent surgery (particularly early surgery) often had stronger surgical indications, representing a higher-risk population. Survivor bias is another concern. Patients who survived long enough to receive surgery might inherently have had better prognoses, which could have influenced the observed outcomes. The retrospective nature of many included studies may have led to incomplete data collection and reporting, further complicating the analysis. This is particularly relevant when assessing the primary outcome of mortality and the secondary outcome of waiting times for surgery. The variability in reporting NCs and the lack of standardized definitions across studies also pose challenges, as these factors can introduce inconsistencies and reduce the comparability of the included studies. Since no study included in the meta-analysis was specifically designed to evaluate the impact of NCs in patients with IE in TAVI, it was not possible to perform subgroup analyses regarding this specific population that is now rapidly growing due to population aging. Furthermore, the analysis may be limited by the lack of individual patient data, which restricts the ability to perform more granular subgroup analyses and adjust for patient-level covariates. Future research should aim to address these limitations by incorporating individual patient data and employing more rigorous and standardized methodologies to enhance the validity and reliability of the findings.

## 5. Conclusions

Our meta-analysis specifically investigated the impact of neurological complications on the outcomes of patients with IE, with a primary focus on all-cause mortality and a secondary focus on the impact of the NC on the timing to cardiac surgery. Neurological complications during IE were significantly associated with increased all-cause mortality. This association was especially pronounced for major NCs, such as ischaemic stroke with severe functional impairment and haemorrhagic stroke, while minor or asymptomatic strokes did not show a significant association with mortality. Regarding the timing of surgery, we did not find a statistically significant delay for patients with NCs compared to those without.

Prospective studies designed to evaluate the optimal timing of surgery for these patients, considering the severity and type of neurological event and incorporating standardized definitions and imaging protocols, are still needed.

## Figures and Tables

**Figure 1 jcm-13-07053-f001:**
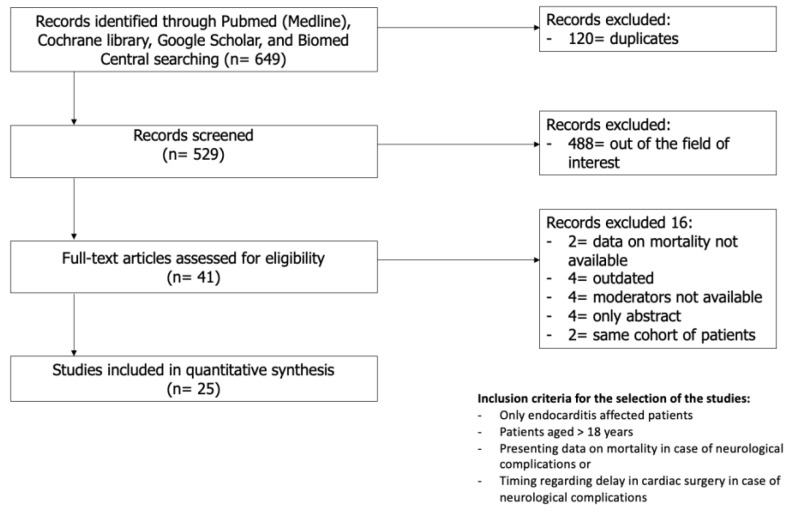
Search strategy.

**Figure 2 jcm-13-07053-f002:**
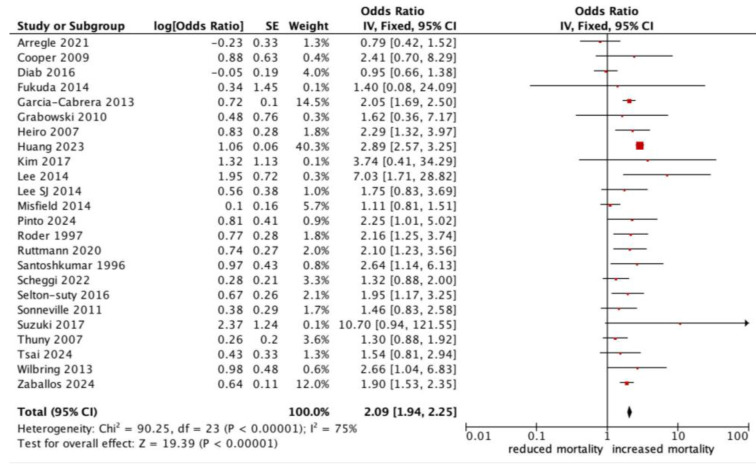
Forest plot for the primary outcome: neurological complications and all-cause mortality.

**Figure 3 jcm-13-07053-f003:**
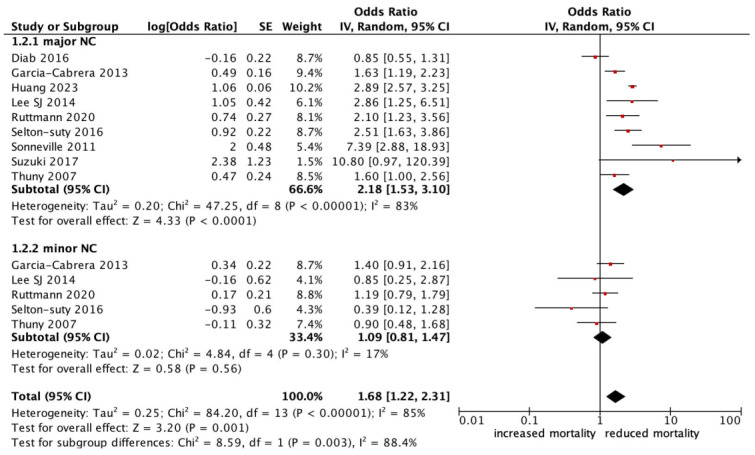
Forest plot for the primary outcomes according to subgroup analysis of major vs. minor neurological complications.

**Table 1 jcm-13-07053-t001:** Studies included in the meta-analysis.

References	Study Design	Pts	NCs	Age (Years) Mean	Male%	Cardiac Surgery NCs/No NCs (n, %)	MV/AV Involvement NCs Patient	PV Involvement	BC + for S.A. NCs vs. No NCs	Right Sided Endocarditis in NCs	HF % in NCs/No NCs	Abscess or Valvular Complications % in NCs	NC Subtypes
Arregle et al., 2021 [16]	P	351	94	68	68%	59 (62.8%) vs. 118 (45.9%)	43 (45.7%)/46(48.9%)	27 (28.7%)	26 (27.7%)/ 45 (17.5%)	.	32 (34%)/ 89 (34.6%)	31.5%	IS 53 SCE 31 IE 38 BA 7
Cooper et al., 2009 [5]	P	56	14	58	64%	.	43%/21%	4 (29%)	7 (50%)/ 19(45%)	.	.	43%	IS 13 (25%) SCE19 (33.9%)
Diab et al., 2016 [17]	R	308	87	62	72%	100% in both cases	36%/46%	12 (13.8%)	38 (43.7%)/ 38 (17.2%)	.	38 (42.7%)/ 124 (56%)	31%	IS 68 (78%) IE19 (22%)
Fukuda et al., 2014 [18]	R	38	16	52	68.4%	100% in both cases	69.5%/12.5%	2 (12.5%)	4 (25%)/ 1 (4.5%)	6%	7 (43%)/ 9 (40.9%)	.	IS 9 (56%) BA 1 (6.3%) MG 1 (6.3%) IE 1 (6.3%) MA 1 (6.3%) IS + MA 3 (18.8%)
Garcia-Cabrera et al., 2013 [19]	R	1345	340	61	68%	32%vs41%	49%/51	47 (14%)	52 (20%)/NR	.	.	.	MG 86 IS 192 IE 60
Grabowski et al., 2011 [20]	R	65	37	46	54%	.	31%/38%	14 (37.8%)	11 (36.6%)/ 10 (45.5%)	.	.	.	IS 13 SCE 24
Heiro et al., 2007 [12]	R	326	86	54.4	71.7%	43% vs. 33.5%	29.4%/34.6%	.	.	5.5%	.	.	NR
Huang et al., 2023 [21]	R	832	112	39.2	63.5%	100% in both cases	71.4%/14.3%	85 (11.8%)	15 (13.4%)/ 61 (8.5%)	.	.	.	IS 240 IE 128 IS + IE 33 MG + BA 15 IS + MG 16 ***
Kim et al., 2017 [22]	R	55	33	54	75%	100% in both cases	66.7%/48.5%	4 (12.1%)	8 (24.2%)/ 5 (22.7%)	6%	.	21%	NR
Lee Su Jin et al., 2014 [23]	R	110	39	52	48%	43.6% vs. 33.8%	38.5%/33.3%	.	.	10.3%	.	.	NR
Lee Seung-Jae et al., 2014 [24]	R	144	37	48	73%	68% vs. 80%	83.8%/45.9%	16 (43.2%)	9 (24.3%)/ 10 (9.3%)	.	.	.	IS 30 (81.1%) TIA 2 (5.4%) IE 16 (43.3%) * MA 1 (3.3%) MG 1 (3.3%)
Misfeld et al., 2014 [25]	R	1571	375	61.8	71%	100% in both cases	36%/44%	90 (24%)	NR	4%	.	.	NR
Okazaky et al., 2013 [26]	R	85	47	62	66%	100% in both cases	40%/47%	9 (19%)	7 (15%)/ 5 (13%)	.	.	.	IS 47 IE 8 (9%) MA 4 (5%) BA 4 (5%)
Pinto et al., 2024 [27]	P	263	48	52	58.3	57% whole cohort	.	10 (20.8%)	14 (29.1%)/ 40 (18.6%)	5.7%	.	29.1%	TIA/IS 60% IE 18% BA/MG 22%
Roder et al., 1997 [28]	R	260	91	68	47	11% vs. 10%	48%/29&	6 (7%)	.	2%	.	.	NR
Ruttmann E. et al., 2020 [14]	P	440	135	54	69%	100% in both cases	47.4%/37.8%	25 (18.5%)	74 (54.8%)/ 154 (50.5%)	0%	.	28.8%	IS 93 (69.9%) MG16 (11.9%), IE 16 (11.9%) BA 10(7.4%) TIA 4 (0.9%)
Santoshkumar et al., 1996 [29]	R	110	58	24.3	53%	0% in both cases	30%/19.5%	2 (3.8%)	6 (11.5%)/ 11 (19%)	7.5%	.	.	NR
Scheggi et al., 2022 [30]	R	551	126	69	63%	81% vs. 77.4%	47.6%/50.8%	50 (39.7%)	34 (27%)/ 69 (16%)	1.6%	.	20.6%	IS 92 (73%) IE 34 (27%)
Selton-Suty et al., 2016 [31]	R	283	135	61	78.5%	51.8% vs. 54%	68.1%/49.6%	38 (28.1%)	43 (31.9%)/ 33 (22.3%)	.	46 (34.6%)/ 35 (23.8%)	23.7%	IS 98 (72.5%) TIA 9 (6.6%) IE 33 (24.4%) MG 17 (12.5%) BA 13 (9.6%) MA 9 (6.6%)
Sonneville et al., 2011 [32]	P	198	108	61	66%	51.6% vs. 48.8%	58%/25%	17 (16%)	60 (56%)/ 31 (34%)	.	16 (15%)/ 20 (22%)	12%	IS 79 ** IE 53 MG 41 BA 14 MA 10
Suzuki et al., 2017 [33]	R	68	25	63	48%	76% vs. 58%	60%/36%	13 (52%)	6 (24%)/ 2 (5%)	.	6 (24%)/ 5 (12%)	28%	IS 25 (100%) IE 7 (28%)
Thuny et al., 2007 [34]	P	496	109	59	74%	58% vs. 60%	58%/56%	24 (22%)	31 (28%)/ 68 (18%)	.	31 (28%)/ 148 (38%)	33%	SCE 17 IS 43 IS + IE 7 IE 12 TIA 30
Tsai et al., 2024 [35]	R	392	92	52.3	66.3%	100% in both cases	71.6%/40.3%	.	.	.	.	3.6%	IS 65 IE 16 IS + IE 11
Wilbring et al., 2014 [36]	R	495	70	54	56%	100% vs. not reported	37%/50%	.	40 (57.1%)/ 110 (25.9%)	.	.	.	IS 53 (75.7%) IE 6 (8.6%) MG 9 (12.9%) BA 1.4%
Zaballos et al., 2024 [37]	P	5667	1125	69	63.7%	40.3% vs. 48.4%	54.9%/52.5%	386 (34.3%)	331 (29.4%)/ 941 (20.7%)	2.7%	437 (38.8%)/ 1816 (40%)	35%	IS 818 (72.7%) # TIA 4 IE 127 (11.2%) MA 62 (5.5%) Other 27 (2.4%)

Pts: patients, NCs: neurological complications, MV: mitral valve, AV: aortic valve, PV: prosthetic valve, HF: heart failure, S.A.: *Staphylococcus aureus*, R: retrospective, P: prospective, IS: ischaemic stroke, SCE: silent cerebral embolism, IE: intracranial haemorrhage, BA: brain abscess, MA: mycotic aneurysm, MG: meningitis, TIA: transient ischaemic attack, BC: blood culture, * some IE occurred together with IS. ** 68 patients out of 109 had more than one NC. *** NC subtypes refer to the total cohort of patients.. # 193 of them with haemorrhagic transformation. .: data not available.

**Table 2 jcm-13-07053-t002:** Studies reporting time between neurological complications and cardiac surgery.

Study	Median	Mean	Type of Neurological Complications	Peri-Operative Mortality	Comment
Fukuda et al., 2014 [18]	37 NC/30 no NC	27.8 ± 27.8 solo per NCH	Cerebrovascular infarction 9 (56%) Abscess 1 (6.3%) Meningitis 1 (6.3%) Cerebral haemorrhage 1 (6.3%) Mycotic aneurysm 1 (6.3%) Infarction + mycotic aneurysm 3 (18.8%)	1 (6.3%)	Trial dedicated to diagnostic algorithm to minimize unnecessary diagnostic tests and achieve good outcomes for these challenging patients.
Ruttmann E. et al., 2020 [14]	4 (0–38) NC/8 (0–90) no NC	4 ± 6/8 ± 15	Cerebrovascular stroke only 93 (69.9%) 42 (31.1%) had a complicated stroke: additional meningitis 16 (11.9%), haemorrhagic stroke 16 (11.9%), or intracranial abscess 10(7.4%) TIA 4 (0.9%)	17 (12.6%)	Early surgery in patients with IE-related stroke seems to be safe and associated with a very low perioperative cerebral haemorrhage risk. Furthermore, the neurological recovery potential was high among survivors with both patients with uncomplicated ischaemic and complicated stroke.
Cooper et al., 2009 [5]	6 NC/3 no NC (2–8 IQR)	5.5 ± 4/3 ± 1.5	Clinical stroke 14 (25%) Subclinical brain embolization by MRI 19 (33.9%)	43%—30 days mortality	Brain imaging with MRI reveals the presence of SCBE in a substantial proportion of patients with definite left-sided IE, particularly in those with S aureus as the causative organism. Therefore, the overall incidence of ABE appears to be higher than that detected by previous clinical studies
Diab et al., 2016 [17]		11.8 ± 17.6/10.9 ± 15	Ischaemic stroke 68 (78%) Haemorrhagic stroke 19 (22%)	27.2%	Pre-operative stroke and neurological disability do not independently affect short- and long-term mortality in patients with infective endocarditis. It appears that patients with pre-operative stroke present with a generally higher risk profile.
Okazaki et al., 2013 [26]		22 ± 27/22 ± 27	Acute ischaemic lesion 47, of them 28 (60%) with small lesions, 36 (77%) with multiple lesions. Haemorrhagic lesion 8 (9%) Mycotic aneurysm 4 (5%) Cerebral abscess 4 (5%)	Not reported	Preoperative MRI showed a high incidence of asymptomatic cerebral lesions in definite left-sided IE patients who required cardiac surgery.
Selton-Suty et al., 2016 [31]		16.1 ± 17.1/14.7 ± 19	Symptomatic neurological complication 100 - Ischaemic stroke 68 (69.4%) - TIA 9 - Cerebral haemorrhage 26 - Meningitis 17 - Brain abscess 10 - Mycotic aneurysm 4	14 (20%) Asymptomatic NC 3 (11.1%) Symptomatic NC 42 (25.6%)	Neurological events remain a severe complication of IE when they are symptomatic. However, the use of systematic NIP allows early identification of asymptomatic complications which often prompts practitioners towards a more aggressive management with more frequent surgery, and this was associated with a better prognosis in our study.
Suzuki et al., 2017 [33]		5 ± 10/11 ± 14	Ischaemic stroke 25 complicated by cerebral haemorrhage 7 (28%)	5 (20%)	Early cardiac surgery may provide clinical advantages overcoming peri-operative adverse events in those with IE complicated with cardio-embolic strokes.
Wilbring et al., 2014 [36]		8.7 ± 10.3/Not reported	Ischaemic stroke 75.7% Intracerebral haemorrhage 8.6% Meningoencephalitis 12.9% Intracerebral abscess 1.4% Subarachnoidal bleeding 1.4%	17.2%	NVE complicated by neurological events remains a challenging disease with high mortality and morbidity. Cardiac surgery seemed to be safe in the observed time interval, particularly for patients suffering from ischaemic stroke.
Thuny et al., 2007 [34]	9 (0–2146)	541 ± 536 only in NC patients	Symptomatic stroke	4.8%	In patients with IE, mortality and neurologic outcome depend on the type of CVC. Although patients with stroke have a significant excess mortality, particularly in the case of mechanical prosthetic valve IE or an impaired consciousness, those with silent CVC or TIA have a relatively good prognosis. Even if valvular surgery can exacerbate cerebral damages after CVC, the risk of postoperative neurologic exacerbation seems to be low after a silent CVC, TIA, and non-massive ischaemic stroke
Huang et al., 2023 [21]		1.95 ± 0.08 vs. 2.25 ± 0.06 months	Defined as symptomatic neurological complications 100%	14.3%	Symptomatic neurological complications before surgery are associated with higher in-hospital mortality following cardiac surgery and prolonged intubation time.
Sonneville et al., 2011 [32]	10 (1–19) only in NC patients	10 ± 5/NR	Ischaemic stroke 79 Intracranial haemorrhage 53 Meningitis 41 Brain abscess 14 Mycotic aneurysm 10	NR	Neurologic events are the most frequent complications in IE patients requiring ICU admission. They contribute to a severe prognosis, leaving less than one-third of patients alive with functional independence. Neurologic failure at ICU admission represents a major determinant of mortality regardless of the underlying neurologic complication.
Tsai et al., 2024 [35]	7 (3–11) vs. 6 (3–12)	7 ± 2/6.75 ± 2.25	Ischaemic stroke 70.6% Intracerebral haemorrhage 17.3% Both 12%	18.5%	Neurologic complications should not delay the timing of surgical interventions. Early cardiac surgery may be associated with more favourable clinical outcomes in patients with such neurologic complications.

NCs: neurological complications, TIA: transient ischaemic attack, NR: not reported, NVE: native valve endocarditis, IE: infective endocarditis; ICU: intensive care unit; CVC: cerebrovascular complication; MRI: magnetic resonance imaging; ABE: acute brain embolization; SCBE: subclinical brain embolization.

## Data Availability

The authors confirm that the data supporting the findings of this study are available within the article and its Appendix A.

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
