# Peer review of "The Impact of Neurological Complications in Endocarditis: A Systematic Review and Meta-Analysis"

_jcm, 2024, doi:10.3390/jcm13237053_

Round 1
Reviewer 1 Report
Comments and Suggestions for Authors
Thank you for the opportunity to review this manuscript. This is a systematic review and meta-analysis of neurologic effects outcomes in those with infective endocarditis. The systematic review is stronger than the meta-analysis for several reasons. 1) there is a large number of studies with vast heterogeneity which the authors attempted to synthesize. 2) there is inconsistency in reporting the aims and statistical results; some due to a lack of consistent variable measurement. With some revision, however there are issues to be addressed first.
The authors point to the risk of methodological heterogeneity among studies as a study limitation, and I would go further to say that this risk is an issue in this case for meta-analyses of these individual studies. “the definition of major versus minor stroke varied across studies.” (line 302) “mean surgical waiting times varied greatly between the included studies.” (lines 333-334)
How was neurological deficit or disability measured to determine whether there existed minor or major NC on data extraction? Please describe this process in section 2.2. What is “limited residual neurological deficit as opposed to “significant neurological disability”? (lines 118-123)
In addition, the authors draw conclusion of causation from the results which need to be scaled back to an association inferred from odds ratio (lines 385-386; 257-258). Were the factors listed in lines 255 and 256 predictors or were they correlated? Also, in lines 71-74, the aims are restated to determine impact rather than association. This is different and needs to be consistent.
Please clarify the stated study objective in lines 337-340.
Please cite the study or studies reporting prevalence for mitral valve endocarditis in lines 266-268.
Please clarify the sentence in line 235-237. I think the “not” in “demonstrate a [not] significant standardized mean difference” should be removed.
Please clarify in Figure 1 Search strategy – In the bottom left box, should this be quantitative synthesis, rather than qualitative synthesis?
In line 152, please be more specific with “drug users” – do you mean IV drug users, those using “dirty needles” ………….
Line 106 – typo MC to NC.
Author Response
Comment #1
Thank you for the opportunity to review this manuscript. This is a systematic review and meta-analysis of neurologic effects outcomes in those with infective endocarditis. The systematic review is stronger than the meta-analysis for several reasons. 1) there is a large number of studies with vast heterogeneity which the authors attempted to synthesize. 2) there is inconsistency in reporting the aims and statistical results; some due to a lack of consistent variable measurement. With some revision, however there are issues to be addressed first.
Reply #1
We would like to thank the Reviewer #1 for taking into consideration our manuscript. We apologize for being not sufficiently clear in the first version of the manuscript. We prepared a revised version of the paper, addressing all questions raised by the Reviewer#1.
Comment #2
The authors point to the risk of methodological heterogeneity among studies as a study limitation, and I would go further to say that this risk is an issue in this case for meta-analyses of these individual studies “the definition of major versus minor stroke varied across studies.” (line 302) “mean surgical waiting times varied greatly between the included studies.” (lines 333-334)
Reply #2
We really agree with the Reviewer about the risk of heterogeneity in the meta-analysis. Therefore as suggested we better specified which are sources of heterogeneity.
Modified text: Section “discussion” lines 367-370: and regarding the outcomes and analysis performed in particular, as already pointed out a great source of heterogeneity is present in the definition of major versus minor NC as well as in mean surgical waiting times across studies included.
Comment #3
How was neurological deficit or disability measured to determine whether there existed minor or major NC on data extraction? Please describe this process in section 2.2.
Reply #3
We thank the Reviewer for this comment. We have performed data regarding subgroup analysis using the OR for minor or major NC already present in the published studies included in the meta-analysis. In each study included in this specific subgroup analysis NC were classified as minor or major according to definition present in each single study and reported in Table 2s. Of course definitions are different among studies. For being more clear we added a column in table 2s where we specified what we used for minor or major NC.
Modified text: Table 2s
Modified text: In Table 2s are described the definitions used by each single study to determine major or minor NC.
Comment #4
What is “limited residual neurological deficit as opposed to “significant neurological disability”? (lines 118-123).
Reply #4
We are sorry if we were not clear enough. In minor NC the residual neurological deficit has to be limited, while in major NC the residual neurological deficit has to be significant and related to disability.
Modified text: Section “methods”, lines 121-123:
Minor NC was defined as a minor stroke with limited residual neurological deficit, transient ischaemic accident or asymptomatic stroke and a major NC defined as a stroke leading to significant neurological deficit and disability,territorial stroke or haemorrhagic stroke.
Comment #5
In addition, the authors draw conclusion of causation from the results which need to be scaled back to an association inferred from odds ratio (lines 385-386; 257-258).
Reply #5
We really thank the Reviewer for this comment. We have amended the text as suggested.
Modified text: Section “Discussion”, lines 257-260
- Major stroke or haemorrhagic stroke were related to all-cause mortality, whereas minor strokes/TIA or asymptomatic strokes did not.
- The presence of a NC was not significantly associated to the time to surgery among patients with and without NC.
Modified text: Section “conclusions”, lines 393-397
Neurological complications during IE were significantly associated to increased all-cause mortality. This association was especially pronounced for major NC, such as ischemic stroke with severe functional impairment and haemorrhagic stroke, while minor or asymptomatic strokes did not show a significant association with mortality.
Comment #6
Were the factors listed in lines 255 and 256 predictors or were they correlated? Also, in lines 71-74, the aims are restated to determine impact rather than association. This is different and needs to be consistent.
Reply #6
We really thank the Reviewer for the comment, we have amended the text accordingly.
Modified text: Section “Discussion”, lines 257-260
- Major stroke or haemorrhagic stroke were related to all-cause mortality, whereas minor strokes/TIA or asymptomatic strokes did not.
- The presence of a NC was not significantly associated to the time to surgery among patients with and without NC.
Modified text: Section “introduction”, lines 71-74: i) evaluate the association between NC in patients with IE and mortality focusing particularly in those with minor or asymptomatic strokes; and ii) determine if the occurrence of NC in patients with IE is associated to delay in the timing to surgical intervention.
Comment#7
Please clarify the stated study objective in lines 337-340.
Reply#7
We are sorry for been not enough clear. We rephrase the sentence.
Modified text: Section “discussion” lines 340-342:
In our study, we showed that there was no significant difference in the mean-time to surgery in patients with infective endocarditis complicated by a NC compared to those without NC.
Comment #8
Please cite the study or studies reporting prevalence for mitral valve endocarditis in lines 266-268.
Reply # 8
We are sorry for the imprecision, the references for this sentence are:
- Das AS, McKeown M, Jordan SA, Li K, Regenhardt RW, Feske SK. Neurological Complications and Clinical Outcomes of Infective Endocarditis. J Stroke Cerebrovasc Dis. 2022 Aug;31(8):106626. doi: 10.1016/j.jstrokecerebrovasdis.2022.106626.
- Yanagawa B, Pettersson GB, Habib G, Ruel M, Saposnik G, Latter DA, Verma S. Surgical Management of Infective Endocarditis Complicated by Embolic Stroke: Practical Recommendations for Clinicians. Circulation. 2016 Oct 25;134(17):1280-1292. doi: 10.1161/CIRCULATIONAHA.116.024156.
We have added the references
Modified text: References added (37,38)
Comment #9
Please clarify the sentence in line 235-237. I think the “not” in “demonstrate a [not] significant standardized mean difference” should be removed.
Reply #9
We are sorry for the typo. We amended the text.
Modified text: removed [not]
Comment #10
Please clarify in Figure 1 Search strategy – In the bottom left box, should this be quantitative synthesis, rather than qualitative synthesis?
Reply #10
We are sorry for the typo. The correct word was “quantitative”.
Modified text: Figure 1
Comment #11
In line 152, please be more specific with “drug users” – do you mean IV drug users, those using “dirty needles”
Reply#11
Sorry for the imprecision, we added “intravenous” to drug users.
Modified text: line 153: intravenous drug users
Comment #12
Line 106 – typo MC to NC.
Reply#12
We are sorry for the typo, we amended the text.
Modified text: line 106: NC

Reviewer 2 Report
Comments and Suggestions for Authors
Thank you for the possibility to read and review this interesting paper by Federico Sanguettoli concerning the implications ofmplications in the neurological co course of endocarditis.
The study was well-designed, and the paper is presented in an interesting and detailed manner. The introduction presents the main problems related to neurological complications in IE patients. The methods section is precisely described. Discussion relates both to the results and explanations of the mentioned issues and problems.
However, in current TAVI progress, the manuscript lacks the important issue of neurological complications in this particular group of patients. As the population is growing very rapidly, it should not be missed. Please complete the paper by adding this important issue.
Author Response
Comment #1
Thank you for the possibility to read and review this interesting paper by Federico Sanguettoli concerning the implications of complications in the neurological course of endocarditis.
The study was well-designed, and the paper is presented in an interesting and detailed manner. The introduction presents the main problems related to neurological complications in IE patients. The methods section is precisely described. Discussion relates both to the results and explanations of the mentioned issues and problems.
Reply#1
We really thank the Reviewer #2 for the positive comment.
Comment #2
However, in current TAVI progress, the manuscript lacks the important issue of neurological complications in this particular group of patients. As the population is growing very rapidly, it should not be missed. Please complete the paper by adding this important issue.
Reply #2
We thank the Reviewer for this comment. However, in the study included in the meta-analysis there were not study specifically intended to transcatheter aortic valve replacement (TAVI). Therefore we were not able to perform a subanalysis regarding this specific population, even though this is now a rapidly growing groups of patients do to population aging, we added this note in the paragraph “study limitation”.
Modified text: Section “discussion”, lines 381-384
Since no study included in the meta-analysis was specifically designed to evaluate the impact of NC in patients with IE in TAVI, it was not possible to perform subgroup analyses regarding this specific population that is now rapidly growing do to population aging.

Reviewer 3 Report
Comments and Suggestions for Authors
This is an important topic. The search strategy with MeSH terms and exclusion criteria is present. There is a clear reasoning. The first table is not easy to read and should be restructured.
Minor issue: there are some typo's needing correction

Author Response
Comment #1
This is an important topic. The search strategy with MeSH terms and exclusion criteria is present. There is a clear reasoning.
Reply#1
We really thank the Reviewer #3 for the positive comment.
Comment #2
The first table is not easy to read and should be restructured.
Minor issue: there are some typo's needing correction.
Reply #2
We agree with the Reviewer#2 that Table 1 is a bit larger, we tried to modified the layout of the table to improve its appearance. Finally we have re-reviewed the text removing typos. Thank you for your advices.
Modified Text: Table 1: line spacing and abbreviations in the Table.

Reviewer 4 Report
Comments and Suggestions for Authors
We would like to express our gratitude to the authors for submitting their manuscript to our journal. The article presents a systematic review and meta-analysis examining the impact of neurological complications within the context of infective endocarditis. After careful consideration, we recommend the following minor revisions to enhance the clarity and depth of the study:
1) We advise the authors to create a graphical abstract summarizing the key messages and main findings of the study. This visual representation will aid in conveying the essential points to readers and enhance the manuscript's accessibility.
2) It is recommended that the authors cite the article by Pizzino et al. (PMID: 38786960, PMCID: PMC11121817, DOI: 10.3390/jcdd11050138). This reference discusses patients with infective endocarditis undergoing surgery and demonstrates that embolic events and heart failure associated with infection are significant negative prognostic factors.
3) Finally, the authors should consider discussing any gender-related differences regarding neurological complications in patients suffering from infective endocarditis. This aspect could provide valuable insights and contribute to a more comprehensive understanding of the subject.
We look forward to the authors’ revisions and appreciate their effort in contributing to this important area of research.
Author Response
Comment#1
We would like to express our gratitude to the authors for submitting their manuscript to our journal. The article presents a systematic review and meta-analysis examining the impact of neurological complications within the context of infective endocarditis. After careful consideration, we recommend the following minor revisions to enhance the clarity and depth of the study:
1)We advise the authors to create a graphical abstract summarizing the key messages and main findings of the study. This visual representation will aid in conveying the essential points to readers and enhance the manuscript's accessibility.
Reply#1
We thank the Reviewer for the comment. The graphical abstract was already uploaded with the manuscript file, we will upload it once again if the process had failed. Thank you for your suggestion.
Comment #2
2) It is recommended that the authors cite the article by Pizzino et al. (PMID: 38786960, PMCID: PMC11121817, DOI: 10.3390/jcdd11050138). This reference discusses patients with infective endocarditis undergoing surgery and demonstrates that embolic events and heart failure associated with infection are significant negative prognostic factors.
Reply#2
We thank the Reviewer for the suggestion. We have added the suggested reference.
Modified text: reference added (4)
Comment #3
3) Finally, the authors should consider discussing any gender-related differences regarding neurological complications in patients suffering from infective endocarditis. This aspect could provide valuable insights and contribute to a more comprehensive understanding of the subject.
We look forward to the authors’ revisions and appreciate their effort in contributing to this important area of research.
Reply #3
We really thank the reviewer for this important comment. However, as showed by results from the meta-regression analysis regarding the primary endpoint of the meta-analysis, the sex (expressed as percentage of male per each study) did not influence the mortality in patients with neurological complication and infective endocarditis (β = -0.01, p = 0.450)(see supplementary material Table 3s).

Round 2
Reviewer 2 Report
Comments and Suggestions for Authors
Thank you for claryfication